# A New Way of Toughening of Thermoset by Dual-Cured Thermoplastic/Thermosetting Blend

**DOI:** 10.3390/ma12030548

**Published:** 2019-02-12

**Authors:** Shankar P. Khatiwada, Uwe Gohs, Ralf Lach, Gert Heinrich, Rameshwar Adhikari

**Affiliations:** 1Research Center for Applied Science and Technology, Tribhuvan University, Kiritipur, Kathmandu 44613, Nepal; skradius33@gmail.com (S.P.K.); nepalpolymer@yahoo.com (R.A.); 2Leibniz-Institut für Polymerforschung Dresden e.V., Hohe Straße 6, 01069 Dresden, Germany; gheinrich@ipfdd.de; 3Polymer Service GmbH Merseburg, Eberhard-Leibnitz-Straße 2, 06217 Merseburg, Germany; ralf.lach@psm-merseburg.de; 4Institut für Textilmaschinen und Textile Hochleistungswerkstofftechnik, Technische Universität Dresden, 01069 Dresden, Germany

**Keywords:** block copolymers, dual curing, electron beam, epoxy resins, toughness

## Abstract

The work aims at establishing the optimum conditions for dual thermal and electron beam curing of thermosetting systems modified by styrene/butadiene (SB)-based triblock copolymers in order to develop transparent and toughened materials. The work also investigates the effects of curing procedures on the ultimate phase morphology and mechanical properties of these thermoset–SB copolymer blends. It was found that at least 46 mol% of the epoxidation degree of the SB copolymer was needed to enable the miscibility of the modified block copolymer into the epoxy resin. Hence, an electron beam curing dose of ~50 kGy was needed to ensure the formation of micro- and nanostructured transparent blends. The micro- and nanophase-separated thermosets obtained were analyzed by optical as well as scanning and transmission electron microscopy. The mechanical properties of the blends were enhanced as shown by their impact strengths, indentation, hardness, and fracture toughness analyses, whereby the toughness values were found to mainly depend on the dose. Thus, we have developed a new route for designing dual-cured toughened micro- and nanostructured transparent epoxy thermosets with enhanced fracture toughness.

## 1. Introduction

### 1.1. General Remarks

Thermosetting materials are used as high-performance materials such as adhesives, composites, and coatings in the aerospace and electronics industries [1,2,3,4]. However, the high performance of the material is limited by its brittle behavior because of its poor resistance to crack initiation and low fracture energy value [5]. Ineffective compatibilization of thermosetting mixtures of polymers with high interfacial tension often presents poor mechanical properties for the blends. So, for high-performance applications, the toughness of the epoxy thermosets should be increased by improving their resistance to crack propagation without significant drops in other important inherent properties such as modulus, glass transition temperature (T_g_), and transparency. To overcome this limitation, many attempts have been successfully completed for thermally cured systems by incorporating the second phase into the matrix of epoxy resins through physical blending or chemical reactions [6,7,8,9,10].

Among some established toughening strategies, the molecular modifications induced by the electron beam (EB) irradiation strategy can strongly alter the mechanical, electrical, and thermal properties of the polymers [11,12,13]. Therefore, the EB curing is considered as an effective and sustainable method for modification of polymers in comparison to thermal curing [14,15]. The use of EB-modified materials for transport applications has considerably increased in the last few decades for their favorable specific strength and resistance to both corrosion and chemicals as well. The benefits of EB curing for the manufacturing of high-performance materials have been widely discussed, and the importance of this process was suggested by different research groups [12,13,16].

EB curing was successfully tested on acrylic derivatives of epoxies, leading to polymerization via a radical mechanism. However, the final product did not address the required glass transition temperature (T_g_), elastic modulus, and fracture toughness for aerospace and advanced automotive applications [15,17,18,19]. Nevertheless, the enhanced thermal and mechanical properties of the EB-cured materials similar to thermal curing were obtained by the cationic polymerization of epoxy resins using suitable onium salts as initiators [20]. Several parameters, such as composition and structure of the epoxy resin, dose, dose rate, gas atmosphere, and curing temperature, can greatly influence the final properties of the cured materials [21,22]. Epoxy thermoset is prepared by either thermal curing or EB curing. Both curing methods provide high glass transition temperatures and high elastic moduli. EB curing uses accelerated electrons in order to polymerize and crosslink the resins.

### 1.2. The EB Irradiation Process

EB irradiation is a sustainable technique and has been introduced in the field of material science nearly fifty years ago. The earliest use of this technology was for the sterilization of disposable medical products, conservation of food products, and crosslinking of plastic materials. The curing processes of resin-based materials were developed a bit later [23]. In recent years, the application of this technology has grown more and more. It is widely used to produce heat-shrinkable plastic films, as insulator for electrical wires, and cables in order to increase heat resistance and to enhance the resistance against abrasion and solvents. Besides these applications, EB irradiation can induce the reduction of the molecular mass by scission of the polymers [17]. This leads to the formation of highly applicable engineering polymeric materials which are extensively used in the automotive and aeronautic/aerospace industries [15,18,19,24].

The EB irradiation can be used to produce micro- and nanostructure systems [25]. Accelerated electrons gradually lose their energy through a huge number of small energy transfers. Basically, the effect of EB irradiation depends on several parameters. These are the absorbed dose (D); the energy absorbed per unit of mass, measured in Gray (Gy)), dose rate (energy absorbed per unit of mass and unit time, measured in Gy/s) [26], gas atmosphere, and the constitution of the polymer to be irradiated. The interaction of EB with the polymer causes the formation of excited states or ions and secondary electrons (Scheme 1). These reactive species are converted into free radicals, leading to chain scission, molecular degradation, chain branching, and cross-linking [17]. The ratio of crosslinking to scission reaction depends on the constitution of polymers, as well as the EB treatment conditions. The outlines of the reaction mechanisms are shown in Scheme 1 [27].

The irradiation process allows for the curing reactions; in some cases, post-thermal treatments have been subjected in order to complete the curing reaction. It is important to consider that the post-thermal curing is performed on already polymerized materials. The combination of a dual curing process, i.e., radiation curing at moderate temperature, followed by a thermal treatment at high temperature, allows for an improvement of the thermal performance, and significant increase of the elastic modulus. However, the low-fracture toughness originates from the highly crosslinked structure restricting the molecular behavior and can give rise to very brittle materials. Therefore, these materials need to be toughened to enable more applications. The toughness of EB-cured epoxy thermoset can be increased by the use of suitable toughening agents [6,7,8,9,10,13]. A post-irradiation thermal curing allows us to uniform the structure and to obtain a sufficiently high value of T_g_, indicating an increase of the crosslinking degree. The presence of the thermoplastic elastomer as toughening agent induces a marked effect on toughness.

EB-cured epoxy resin (80–90 kGy) containing 10 wt.% polyether sulphone (PES) has a value of the elastic modulus of about 3.6 GPa. The critical intensity factors K_IC_, i.e., the fracture toughness for both diglycidyl ether of bisphenol F (DGEBF) neat resin and DGEBF/PES blends, are reported [28,29]. In addition, the K_IC_ values and the related references of epoxy resins are summarized in the comprehensive data compilation given by Lach and Grellmann [30]. When passing from neat epoxy resin systems to blends, increased K_IC_ values were observed [31].

Recent trends in the global situation towards environmentally conscious manufacturing practices have led us to look for alternative thermal curing technologies. The EB curing is considered as an effective and appropriate method for the modification of polymers in comparison to thermal curing because of its fast curing time, ambient curing temperature, homogeneity, and thermal initiator free curing [14,15]. This modern approach is generally referred to as the green method.

### 1.3. Motivation

In this paper we are looking for a new strategy to make a toughened and transparent epoxy thermoset for advanced applications. If the epoxy resin was modified by a polymer containing a high amount of polybutadiene (PB), the blend is not transparent and toughened. Here a new approach is described to increase toughness and maintain stiffness and transparency by EB irradiation followed by thermal curing. Finally, the toughened materials are prepared by using small amounts of block copolymers (BCPs), so this method is cost effective as well.

Numerous studies have been carried out to investigate the effects of block copolymer on the mechanical, thermal, and electrical properties of modified epoxy systems [18,19,24,32,33,34,35], but the details on the effects of the dose on the ultimate phase morphology and mechanical properties of the dual-cured epoxy thermosetting system modified by block copolymers were not reported. Hence, the aim of this research was to achieve the preparation of toughened dual-cured micro- and nanostructured transparent epoxy thermosets. The ultimate phase morphology and thermo-mechanical properties were analyzed. This research paper also summarizes the important aspects of EB-cured epoxies and their processes.

## 2. Experimental Details

### 2.1. Materials and Specimen Preparation

Diglycidyl ether of bisphenol A, provided by Sigma Aldrich, was used as the matrix (Figure 1a), and 4,4-diaminodiphenylmethane (DDM), purchased from Sigma Aldrich, was used as the curing agent (Figure 1c). The commercial styrene/butadiene-based linear triblock copolymer named as Kraton D1101 (31% by volume of styrene) was used as a modifier [34] (Figure 1b). The lab grade chemicals were used without any further purification.

#### 2.1.1. Epoxidation of Kraton D1101 by *m*-CPBA

The epoxidation of the styrene/butadiene-based triblock copolymer was accomplished by using m-chloroperoxybenzoic acid (*m*-CPBA) [35]. In a typical procedure, 100 g of Kraton D1101 and 500 mL of dichloromethane were poured into a three-necked round bottom flask and stirred until the polymer was completely dissolved. To achieve a degree of epoxidation (DOE) of around 50 mol%, stoichiometric amounts of *m*-CPBA were charged to the polymer solution. The mixture was then vigorously agitated for two hours under nitrogen gas at 0 °C. After the reaction was completed, the polymer solution was filtered. The excess of *m*-CPBA was removed by extraction with a saturated aqueous sodium bicarbonate (NaHCO_3_) solution and the mixture was further dried by the sodium sulphate (Na_2_SO_4_) solution. The final polymer solution was recovered using a separating funnel and a vacuum suction pump. The solvent was completely evaporated to recover the solid residue which was dried under a vacuum at room temperature. The chemical reaction involved herein is shown in Figure 2. The epoxidized Kraton D1101 is denoted as eKraton.

The oxygen-oxygen bond in *m*-CPBA is quite weak (about 33 kcal/mol) which leads to the highly reactive nature. One of the prime impacts of this convention is that the stereochemistry is always retained. That means, a cis-alkene always gives the cis-epoxide, and a trans-alkene gives a trans-epoxide product. This is a prime example of a stereoselective reaction in organic chemistry. During the reaction, a very reactive transition state is formed, where the bond between the oxygen and the alkene is being formed at the same time that the O–O bond is breaking, and the proton is being transferred from the OH to the carbonyl oxygen. Those little-dotted lines represent partial bonds. The mechanism of *m*-CPBA/BCPs system is shown in Scheme 2. 

#### 2.1.2. Electron Beam Treatment of EP/eKraton Blends

eKraton was dissolved in excess of dichloromethane and mixed with epoxy resin (EP) in the ratio of 90/10 wt.%. The logic behind taking only 10 wt.% of eKraton was to prevent the reaction mixture from gelation before curing. The mixture was then kept in a vacuum oven to allow the solvent to evaporate. Afterwards, an EB treatment was performed at a temperature of 215 °C under a nitrogen atmosphere with doses of 20 kGy and 50 kGy using an ELV-2 electron accelerator from the Budker Institute of Nuclear Physics (Novosibirsk, Russia) using the procedure we previously reported [36] (the doses were designated as “-20” and “-50”, respectively). Then the mixture of EP/eKraton was placed on an electrically-heated plate in an irradiation vessel which was mounted on the conveyor system of the electron accelerator. All treatments were performed with 1.5 MeV at an electron current of 4.0 mA.

#### 2.1.3. Blending of the Epoxidized Mixture

After that, the pure thermoset resin was prepared by the reaction of a stoichiometric amount of EP with DDM (i.e., epoxy resin/DDM in the ratio of 70/30 wt.%). The mixture was pre-cured at 150 °C for 2 h followed by post-curing at 180 °C for another 2 h. The curing of the epoxy system containing 10 wt.% eKraton with doses of 20 kGy and 50 kGy was carried out as described in Reference [36]. In the case of a typical EB and thermal cured system, first the EB-cured EP/eKraton blends were mixed in a stoichiometric amount of DDM and magnetically stirred over a hot plate at 100 °C until the complete dissolution of DDM leading to the formation of a homogenous mixture. The same thermal curing profile has followed, as in the case of pure thermoset resins. Then, each thermally-cured blend was cooled gradually down to room temperature. Finally, the prepared pure epoxy thermoset and the blend were cut into proper sizes according to the requirements as demanded by different characterization techniques.

### 2.2. Characterization Techniques

#### 2.2.1. Molecular Characterization

The ^1^H-NMR spectra (proton nuclear magnetic resonance) were recorded using a Bruker Advance III 500 spectrometer (Bruker, Billerica, MA, USA) operating at 500.13 MHz for ^1^H. The samples were measured at 30 °C in deuterated chloroform (CDCl_3_) which was also used as a reference (δ (^1^H) = 7.26 ppm).

#### 2.2.2. Morphological Characterization

Optical microscopy (OM): EP/eKraton blends were inspected using a Nikon Optiphot-2 optical microscope (Nikon, Minato, Japan) in transmission mode before and after curing. Small droplets of blends were sandwiched between two clean and dry glass coverslips, slightly pressed against each other, and the morphology was observed at room temperature. 

Scanning electron microscopy (SEM): SEM was used for the study of the morphological features and deformation behaviors of the materials. The fracture surface of fractured materials obtained from impact test was sputter coated with an approximately 20–50 nm thin film of gold film. The thin film has two functions: Avoiding surface charging and irradiation damage. The experiments were carried out by using scanning electron microscopy (JSM 6300, JEOL, Akishima, Japan) employing secondary electrons (SE) mode.

Transmission electron microscopy (TEM): TEM was used to investigate the phase morphology of the specimens using a transmission electron microscope LIBRA120 (Carl Zeiss AG, Oberkochen, Germany) with an acceleration voltage of 120 kV. Cured bulk thermoset samples were prepared by using a Leica UCT ultramicrotome equipped with a diamond knife with a cutting edge of 45°. Ultra-thin cross-sections of ~60 nm thickness were cut at −80 °C from the sample with an ultra-microtome EM UC6/EM FC6 (Leica Microsystems, Austria) using a diamond knife and were subsequently transferred to carbon-coated grids. The soft polybutadiene (PB)-rich phase was stained with osmium tetroxide to enhance contrast before the TEM imaging. 

#### 2.2.3. Mechanical Testing

Impact strength: The impact strength of the blends was measured using a Charpy impact tester PSW 4 J (Polymer Service GmbH Merseburg, Merseburg, Germany) according to ISO 179-1. The sample dimensions were 100 mm × 10 mm × 3 mm. The notched specimen would be broken with one blow of the hammer. The Charpy impact strength (in kJ/m^2^) was calculated as the ratio of the fracture energy (in J) and the cross-sectional area (in mm^2^). Five successive measurements of each blend were performed, and their mean average result was taken into account.

Fracture toughness: The fracture toughness testing of the blends was conducted using a Zwick universal testing machine Z2.5 (ZwickRoell GmbH & Co. KG, Ulm, Germany) according to ISO 13586 in three-point bending mode. Single edge-notched bend (SENB) samples having a dimension (L × W × B) of 55 mm × 12 mm × 5.85 mm were prepared. Then, a notch with a depth of a = 5.6 mm was initiated by tapping a fresh razor blade. The critical plane strain energy release rate (G_IC_) was determined from the critical plane strain stress intensity factor (K_IC_) obtained. Care was taken to avoid forming a long crack or breaking the sample. Specimens with long cracks, i.e., a/W exceeding 0.55, were discarded. 

Microindentation: The microhardness measurements were carried out at 23 °C by using FischerscopeH100C recording microhardness tester (Helmut Fischer GmbH, Sindelfingen, Germany) equipped with a pyramidal Vickers diamond indenter which was indented into the sample. This technique comprises the continuous measurement of the load applied by an indenter as a function of its indentation depth with the same speed (50 µm/s) for both loading and unloading. The hardness has been measured in the form of the Martens hardness HM (in N/mm^2^) from the maximum load (in N) and the modulus of elasticity (the indentation modulus E_IT_) has been calculated according to the procedure well-described in ISO 14577. At least five load (F)–indentation depth (h) measurements per material with the maximum load of F_max_ = 1 N were done at different positions on the material plate of approximate area 1 × 1 cm^2^ and thickness around 3 mm and the average value was reported.

#### 2.2.4. Dynamic Mechanical Analysis

Dynamic mechanical analysis (DMA): DMA measurements were carried out on the specimens with dimensions of 30 mm × 10 mm × 3 mm using a DMA Q800 V21.1 rheometer (TA Instruments, New Castle, GB) in torsion mode. The dynamic mechanical properties of samples were measured over a broad temperature range (from −70 to 200 °C) at a frequency of 1 Hz. The temperature dependency of storage modulus (G′), loss modulus (G″) and mechanical loss factor (tan δ = G″/G′) were measured.

## 3. Results and discussion

### 3.1. Quantitative Validation of the Chemical Modification

Figure 3 depicts a region of the ^1^H-NMR spectra of Kraton (bottom) and partially epoxidized Kraton (top). The signals in the 7.2–6.2 ppm region result from the aromatic ring of the styrene units. They remain unchanged in the epoxidation reaction and their intensity can be used as internal intensity reference to compare the spectrum of a non-epoxidized and epoxidized sample. The signals of the olefinic protons of the butadiene segments appear in the 5.8–4.8 ppm region. The copolymerization with butadiene can result in *cis*-and *trans*-1,4-butadiene units, respectively, and in vinylic 1,2-butadiene units. The ^1^H chemical shifts of the corresponding protons are summarized in Table 1.

Whereas the signals of *cis*- (A) and *trans*-1,4-butadiene units (B) almost overlap at about 5.5 and 5.4 ppm, the –CH= (C) and =CH_2_ signals (D) of the vinyl-1,2-butadiene units appear well separated at about 5.6 and 5.0 ppm, respectively. Epoxidation results in decreasing intensity of the *cis*- and *trans*-1,4-butadiene signals, but the signal for the 1,2-butadiene unit of Kraton remains almost unchanged. The epoxidation of vinyl–1,2-butadiene was observed in the ^1^H-NMR spectrum only to a low extent, which could be due to the lower reactivity of vinyl bonds in comparison to *cis* and *trans* double bonds, which is also reflected in the Fourier-transform infrared (FTIR) data.

The epoxide groups in the eKraton result in new signals at about 2.9 and 2.7 ppm, representing the *cis*- (E) and *trans*-epoxy groups (F), respectively. The overall signal patterns in the olefinic and epoxide signal regions strongly depend on the degree of epoxidation (DOE). Adjusting the same region of the aromatic styrene signals to the same intensity, the integral values of the rubbery part before and after epoxidation can be compared. The decrease in the intensity of the peaks corresponding to olefinic protons is related to the DOE.

The *cis*- and *trans*-1,4-butadiene units contain two olefinic protons, but the 1,2-butadiene units contain three olefinic protons. This has to be considered in the calculation. Therefore, the overall integral of the olefinic protons is corrected by half the intensity of the signal group at 5.0 ppm. The following Equation (1) is used to determine the DOE.
(1)$$\mathrm{DOE}=1-\frac{\mathrm{Corrected}\;\mathrm{integral}\;\mathrm{area}\;\mathrm{of}\;\mathrm{olefinic}\;\mathrm{protons}\;\mathrm{after}\;\mathrm{epoxidatio}\;n}{\mathrm{Corrected}\;\mathrm{integral}\;\mathrm{area}\;\mathrm{of}\;\mathrm{olefinic}\;\mathrm{protons}\;\mathrm{before}\;\mathrm{epoxidatio}\;n}\times 100\;\mathrm{mol}\%$$

The calculation is based on the integral values of the spectra (see Figure 3). The corrected integral area of the olefinic protons before epoxidation amounts to 4.175 (3.98 + 0.39 − 0.195). In contrast, the corrected integral area of olefinic protons after epoxidation amounts to 2.255 (2.07 + 0.37 − 0.185). Consequently, the DOE was 46 mol%.

### 3.2. Preliminary Morphological Analysis

The photographs of the epoxy thermosets (EP) and their blends fabricated with 10 wt.-% of eKraton and treated with different doses are presented in Figure 4.

All cured blends were not transparent as targeted. Obviously, the pure EP thermoset was transparent (Figure 4a) because the epoxy system consists of a single phase. After the addition of 10 wt.% eKraton modifiers into the epoxy system (EP/eKraton), the blend turned opaque (Figure 4b), possibly due to the macrophase separation of eKraton in the EP matrix. The use of 10 wt.% eKraton was not enough to make the PB block completely compatible with the epoxy matrix. After an EB treatment with a dose of 20 kGy (EP/eKraton-20), the EB-modified blend became more translucent as compared with non-irradiated EP/eKraton, as shown in Figure 4c.

This implies the formation of micro/nanosized domains of eKraton-20 in the blend. As the EB dose further increases to 50 kGy, the blend was almost transparent, as shown in Figure 4d, suggesting the formation of nanostructures in eKraton-50 along with some microstructures. In the case of transparent micro- and nanostructured blends, the material allows the light to pass because the size of the majority of the domain is less than 700 nm. Similar results were also observed in the epoxy/styrene–butadiene–styrene block copolymer (SBS) system, where epoxidized SBS induced the nanostructure in epoxy resin and the transparency of EP/SBS largely depended on the degree of epoxidation [33,38]. For more information about the morphology, the blends were further analyzed using optical microscopy (OM) and transmission electron microscopy (TEM).

### 3.3. Morphological Behavior

Optical micrographs of the blends of EP/eKraton (90/10 wt.%) before and after curing are displayed in Figure 5. It was observed that the partially epoxidized (46 mol%) ePB subchains of eKratons (dispersed phase) were initially miscible with the epoxy matrix (continuous phase), whereas the immiscible PS subchains were arranged in the micrometer range. The epoxy resin (EP) acts as the selective solvent for the eKratons and it manages the driving force for SA mechanisms. The size of the domains was found to be around 3 µm.

The epoxidized butadiene segments were miscible with the epoxy resin, whereas polystyrene parts as well as non-epoxidized parts were immiscible. The non-epoxidized butadiene (PB) parts formed the shell, and polystyrene (PS) parts may formed the core of the domain, as shown in the schematic enlargement presented in the inset of Figure 5a. However, the microstructure of eKraton/EP after curing was not observed (Figure 5b), because the majority of the microdomains of polystyrene were converted into nanodomains, and the nanostructure cannot be observed by optical imaging.

For detail analysis of the size and distribution of the domain in the nanoscale range, TEM analysis were carried out. The EP/Kraton blend morphology with 10 wt.% of unmodified block copolymer is presented in Figure 6a, whereby the morphology of the EP/eKraton blend treated by a different dose is shown in Figure 6b,c. At first glance, heterogeneous morphology was seen in all cases. It is noted that the EP/eKraton blend, which was not cured by EB irradiation, consisting of a bright spherical domain in the 0.5–1 µm range, dispersed in the continuous epoxy matrix, as indicated in light gray. Owing to the electron density difference of different groups and the diversity in the preference of RuO_4_ staining, the continuous areas can be ascribed to the cross-linked epoxy matrix, and the spherical cores as indicated in dark represent the polystyrene parts along with block of non-modified rubbery polybutadiene. The lamellar morphology was also observed inside the spherical core, where light areas correspond to the styrene block, whereas the thin dark shells surrounding the spherical cores can be assigned to the non-epoxidized PB segment.

The average sizes of the spherical microdomains in the EP/eKraton blends were estimated to be 0.5–1 µm using the measuring tool of the software Image J (Laboratory for Optical and Computational Instrumentation (LOCI)). The size of the majority of eKraton domains was larger than the wavelength of light (700 nm). This is the reason why the blend appears opaque, which was in good agreement with the photograph as described in Figure 4b. As the dose increases, the spherical morphology consisting of a bright core and a dark shell remains unchanged; however, the size of the domains were decreased gradually, the number of spherical domains was increased, and the distance between them decreased.

All these results indicate the higher miscibility of eKraton with epoxy resin as the dose of EB treatment increases. Obviously, the EB treatment induced a crosslinking of the ePB subchains with epoxy resin before thermal curing. We conclude that only EB treatment was responsible for the changes in size, positions, and distribution of the domains, since other controlling conditions like post-thermal temperature, hardener, and curing conditions remained constant. In Figure 6b, the sizes of the domains were found to be in the nm−µm range. However, the majority of the domains were in the nm range, and their sizes were comparable with the wavelength of visible light, and hence this blend was translucent in nature, indicating that not all domains were on the nanoscale. These results are in good agreement with the photographs described in Figure 4c. Furthermore, the blends treated by a dose of 50 kGy contain the dispersed domains with sizes in a 50–100 nm scale, as shown in Figure 6c. Since the domain sizes were comparable with the wavelength of visible light, mainly the EP/eKraton-50 blends were transparent. This result is also supported by the image of the micrograph as shown in Figure 4d.

### 3.4. Dynamic Mechanical Behavior

Figure 7 shows the temperature dependencies of the dynamic mechanical properties (storage modulus (G′) and damping coefficient (tan δ)) of the pure epoxy thermoset and EP/eKraton blends with different doses. In general, the addition of modified block copolymers (eKraton) diminishes the storage modulus of epoxy resin due to the presence of a soft rubbery phase at high temperatures (T). Here, the storage modulus for EP/eKraton treated by EB increases for T < 100 °C and decreases for T > 100 °C. The values obtained from DMA are listed in Table 2. The unmodified epoxy has a main glass (α) transition temperature T_g_ = 173 °C and a secondary (β transition) temperature of −40 °C [39,40].

All blends fabricated with 10 wt.% of eKraton with different doses indicated the two transitions (α and β) with slight changes in the storage modulus (G′) and the peaks of the mechanical loss factor (tan δ). Apart from glass transition temperature, the tan δ peak at T_g_ also provides some information about the structure of the crosslinked thermoset provided by its shape. Specifically, the height of the tan δ peak at T_g_ indicates the extent of the crosslinking. A lower height means a higher crosslink density [41]. Moreover, a broader peak suggests better fracture toughness and a better ability to prevent crack propagation and brittle fracture [42]. 

There is no notable difference on the shape of the tan δ peak at T_g_ among all EP/eKraton samples treated by different doses. Interestingly, tan δ, which is the ratio of the loss modulus to the storage modulus shows, a significant reduction in T_g_ (9 K) from EP to EP/eKraton, and an increase (4–6 K) due to irradiation (Table 2). The storage modulus at 50 °C and the β transition temperature have the same trend as found for T_g_ (Table 2). Similar findings have been also reported by Vignoud [43], as irradiation results in a slight decrease of the T_g_ and storage modulus in the rubbery region, ascribed to crosslinks.

### 3.5. Mechanical Behavior

The mechanical properties of the blends are analyzed by subjecting the blends to undergo high-speed impact tests and low-speed fracture toughness and micro-indentation measurements. In Figure 8, the impact strengths of EP/eKraton blends modified with varied doses compared to that of neat epoxy thermoset are presented. In the case of the blends with eKraton, the impact strength increases with dose amount of eKraton. The impact strength of modified epoxy thermosets increases by 25 % for both EP/eKraton-20 and EP/eKraton-50. However, there is no change in the Charpy impact strength for the unmodified epoxy thermoset EP/eKraton within the experimental uncertainty. The good compatibility and strong interaction between the epoxy matrix and epoxidized polybutadiene (ePB) subchains of eKraton after EB irradiation contributed to the improvement of the impact strengths of the blends. However, the value was kept constant in the case of EP/eKraton. This might be because of heterogeneous distribution of the microsized domain, which may facilitate crack initiation and propagation during sudden impact deformation. The results are in good agreements with the morphology (Figure 4) and TEM analyses (Figure 6).

The Charpy impact tests give an idea about the macroscopic mechanical behavior of the blends, while the micromechanical behavior can be evaluated by fracture toughness measurements. Five successive measurements of each blend were performed, and their mean average result was taken into account. Thus, the latter properties of eKraton-modified epoxy thermosets were analyzed by low-speed fracture toughness in order to determine the critical strain energy release rate (G_IC_) and stress intensity factor (K_IC_). The fracture toughness values (G_IC_ and K_IC_) of the blends modified by different doses containing same amount of eKraton (10 wt.%) were compared with pure epoxy thermoset as shown in Table 3.

The G_IC_ value of the unmodified epoxy resin was 0.68 kJ m^−2^. Compared to neat epoxy, the same G_IC_ value was observed for the EP/eKraton within the experimental uncertainty range. This might be due to the heterogeneous distribution of microsized domains, which may facilitate crack initiation and propagation during fracture deformation. However, the fracture toughness values of modified epoxy blends were higher than those of the neat epoxy blends and they increase gradually with increasing dose levels. These results also support the findings obtained from the impact strength analysis (Figure 8).

The G_IC_ values were increased for EP/eKraton-20 and EP/eKraton-50 thermosets by 1.5 and 2.2 times, respectively, as compared to the fracture energy of neat epoxy thermoset. Compared to the neat epoxy thermoset, the nano- and microsized eKratons were capable of toughening the epoxy matrix more effectively due to the strong interaction between the epoxy matrix and ePB sub-chains of eKraton after EB irradiation, as well as the proper dispersion of the micro/nanodomains and improvement of the fracture toughness by resisting the crack initiation and crack propagation. Moreover, the increased toughness was an advantage of the dual curing of the blends via EB irradiation and post-thermal curing in the presence of the small loading of BCPs with comparison to traditional thermal curing. 

The dose is the controlling parameter for electron induced reactions as well as the size and distribution of the microdomains in the EP/eKraton blends (see Figure 6). Consequently, the crosslinking behavior of Kraton D1101 was studied at a temperature of 215 °C in a nitrogen atmosphere for different dose values (50, 100, 200 kGy) in Reference [36]. In Reference [36], the dominance of the crosslinking reaction over chain scission was confirmed. Based on those results, the dose values of 20 and 50 kGy were precisely selected. The preparation of blends using a dose > 50 kGy were not successful due to the high values of gel content (>89%) of Kraton D1101. The formations of micro- and nanostructures inside epoxy thermosets were also reported to improve the fracture toughness significantly [33,44,45,46,47]. However, the actual mechanism of this toughening is still a subject of debate with respect to whether the modifier or the matrix absorb the most energy during deformation.

Indentation techniques are among the simplest, most reliable, and most efficient methods of testing the surface mechanical properties of thermosetting materials. (Micro)indentation testing has been employed in determining the fracture toughness as well the hardness and elastic modulus of polymeric materials such as micro- and nanostructured thermosets [48,49].

The indentation behavior of the materials can be directly linked to the internal morphologies and deformation behaviors of the materials [49]. In order to determine the effects of the differently-modified block copolymer architectures on the surface hardness of block copolymer/EP, the corresponding values of Martens hardness (HM), maximum indentation depth (h_max_), and indentation modulus (E_IT_) are plotted in Figure 9. A closer look at the curves reveals that the h_max_ values of the blends were not changed within the experimental uncertainty range. Compared to the pure epoxy thermoset, the addition of 10 wt.% of eKraton decreased the HM value independently of the dose. This might be attributable to the soft phase. In the case of E_IT_, the value was decreased by the added eKraton and increased with the dose in order to reach the value of the virgin EP at a dose of 50 kGy. Earlier studies on similar systems show that the mechanisms for epoxies toughened by block copolymers are similar to those of conventional rubber-toughened epoxy, which are crack pinning [50], multiple crazing [51], and debonding [52] mechanisms. Figure 10 shows the fracture surface of epoxy blends with the 10 wt.% block copolymer modified by a dose of 50 kGy (Figure 10b) compared with the brittle surface of the pure epoxy thermoset (Figure 10a). The micrograph reveals the existence of tiny cracks, along the main crack growth direction, and these cracks are pinned by the micro- and nanodomains, as indicated by (A) in Figure 10b.

In Figure 10b, the fracture surface was similar to that of ductile material due to the plasticization effect of epoxidized PB phases [53]. During this type of deformation, the load was transferred more effectively to the nanostructured domains from the crosslinked epoxy phase [54]. Likewise, bifurcations of microcracks were observed which result in an increased surface area of cracks, as visualized in (B) in Figure 10b. Obviously, this may explain the toughness increase of the material. The energy dissipation mechanisms could also be related to the formation of shear deformation of eKraton domains in the micro- and nanodomains of epoxy thermosets by a debonding process, as indicated by (C) in Figure 10b.

### 3.6. The Structure−Properties Correlations Scheme

The concept of creating the micro- and nanostructure in the EP/eKraton blends for the construction of toughened thermosets by dual curing methods has been illustrated in Figure 11. When one of the modified blocks of the segments of eKraton is miscible with the epoxy resin (see Figure 11), even in the curing process, and the other segments of non-epoxidized PB segments and PS are immiscible with the epoxy, the ability of self-assembly for Kraton forms the microdomain of eKraton in the epoxy system before curing. However, micro- and nanodomains were formed after dual curing (EB irradiation and subsequent thermal curing) by a reaction-induced phase separation mechanism. Therefore, the phase size of the polymer blends can be controlled in the order of micro- and nanometers. A summary of the structural properties relationship is also shown in Figure 11.

## 4. Conclusions

The partial epoxidation of a commercial styrene/butadiene-based triblock copolymer Kraton D1101 was achieved and, hence, a dual-cured toughened micro- and nanostructured transparent epoxy thermoset was developed. Some of the important conclusions are given below.
The quantitative validation of the partial epoxidation of Kraton D1101 was achieved using ^1^H-NMR spectroscopy.The experimental condition for the novel dual curing method was established by optimizing the curing condition and deformation behavior.The formation of micro- and nanostructures could be observed, and the effect of dual curing on the ultimate phase morphology was investigated using DMA and TEM analysis.The mechanical properties of the modified blends were improved as per the impact strength, fracture toughness, and micro-indentation analyses. The fracture toughness values were found to mainly depend on the dose.

Therefore, the present study has provided an idea for a new toughening strategy for the construction of toughened micro- and nanostructured thermosets. The new materials provide possible uses of the toughened thermosets for advanced industrial applications.

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
