# Peer review of "A New Way of Toughening of Thermoset by Dual-Cured Thermoplastic/Thermosetting Blend"

_materials, 2019, doi:10.3390/ma12030548_

Round 1
Reviewer 1 Report
This manuscript is well written and deserves publication. There are some minor points which require addressing before publication.
Line 95, the authors state that suitable toughening agents can be used in thermosets. Only one reference is provided here, which is lacking, also can the authors list a few. I am not after a full discussion of these just a little more detail for the reader.
Lines 95-103, can the values of KIC be reported in the text to give some reference to the work about to be done.
The chemical structures provided in the experimental section are not o good quality and need to be improved.
The chemical reaction in Figure 2 shows only one epoxide being formed, in fact the alkene which is epoxidised is the less reactive one while the more reactive (mono-sbustituted) is left alone. Is this accurate? If so can the authors cite some evidence or justification? If it is more of a schematic it would be worth modifying the reaction to show that a mixture of products is formed.
Line 347 a symbol is missing from the brackets after 'tan', actually this is quite common for many of the symbols in the following text, I assume this is an issue when converting to PDF.
Figure 8, can the values expressed in the graph be put numerically over the bars (including error) to allow fro easy interpretation. Also have statistics been done on these samples?
Author Response
This manuscript is well written and deserves publication. There are some minor points which require addressing before publication.
Response: We are delighted to learn that the honorable reviewer has appreciated our work. We would like to thank the reviewer for providing valuable suggestion. Accordingly, we have addressed the comments of the reviewer and modified the manuscript.
Line 95, the authors state that suitable toughening agents can be used in thermosets. Only one reference is provided here, which is lacking, also can the authors list a few. I am not after a full discussion of these just a little more detail for the reader.
Response: We added references 6-10
Lines 95-103, can the values of KIC be reported in the text to give some reference to the work about to be done.
Response: We added following information: The KIC values and the related references of epoxy resins are summarized in the comprehensive data compilation given by Lach and Grellmann [Ref].”
Ref.: Lach, R., Grellmann, W. Fracture mechanical properties. Thermosets and high performance composites, In: Grellmann, W., Seidler S. (eds.), Mechanical and Thermomechanical Properties of Polymers. Landolt-Börnstein, Group VIII Advanced Materials and Technologies, Polymer Solids and Polymer Melts, New Series VIII/6A3, Springer-Verlag, Berlin, 2014, 423–492.
The chemical structures provided in the experimental section are not a good quality and need to be improved.
Response: The authors don’t understand this statement of the reviewer. From the authors point of view the chemical structures are of good quality. Most C and H atoms are generally not designated in organic chemistry.
The chemical reaction in Figure 2 shows only one epoxide being formed, in fact the alkene which is epoxidised is the less reactive one while the more reactive (mono-substituted) is left alone. Is this accurate? If so can the authors cite some evidence or justification? If it is more of a schematic it would be worth modifying the reaction to show that a mixture of products is formed.
Response: Thank you for this comment. We added following information: Justification: The oxygen-oxygen bond in m-CPBA is quite weak (about 33 kcal/mol) which leads to the high reactive nature. One of the prime impacts of this convention is that the stereochemistry is always retained. That means, a cis-alkene always gives the cis-epoxide, and a trans-alkene gives a trans-epoxide products. This is a prime example of a stereo-selective reaction in organic chemistry. During the reaction, a very reactive transition state is formed, where the bond between the oxygen and the alkene is being formed at the same time that the O-O bond is breaking and the proton is being transferred from the OH to the carbonyl oxygen. Those little-dotted lines represent partial bonds. The mechanism of m-CPBA/BCPs system is shown in following scheme 2.
Scheme 2: Mechanism of m-CPBA/BCPs system.
Line 347 a symbol is missing from the brackets after 'tan', actually this is quite common for many of the symbols in the following text, I assume this is an issue when converting to PDF.
Response: The authors agree. Now the symbol is visible.
Figure 8, can the values expressed in the graph be put numerically over the bars (including error) to allow for easy interpretation. Also have statistics been done on these samples?
Response:The values of the Charpy impact strength are now put numerically over the bars. Five successive measurements of each blend were performed and their mean average result was taken into account.

Reviewer 2 Report
This manuscript written by Shankar P. Khatiwada et al reports “A new way of toughening of thermoset by dual cured thermoplastic/thermosetting blend”. This work was conducted for making a toughened and transparent epoxy thermoset for advanced application. The data seems solid, and scientifically sound. The paper clearly indicated that the results obtained in this article can contribute to enhancing of toughening of epoxy/styrene/butadiene(SB)-based triblock copolymers by electron beam curing and thermal curing. However, the authors had better include the following points to make this manuscript more informative to readers. Therefore, this manuscript is recommended for the publication in materials after minor revision as below.
1. The author described that “Moreover, the increased toughness was an advantage of dual toughening of the blends via EB irradiation and post-thermal curing” The EB curing condition such as irradiation dose is important one to enhance toughness of EP/eKraton blends. However, only two kinds of irradiation dose (20 and 50 kGy) were conducted to elucidate the effect of mechanical property of EP/eKraton blends. The reaction degree and crosslinking density of EP/eKraton blends as a function of irradiation dose was significant. However, there was not much explanation about that. So, the author should explain this clearly.
Author Response
This manuscript written by Shankar P. Khatiwada et al reports “A new way of toughening of thermoset by dual cured thermoplastic/thermosetting blend”. This work was conducted for making a toughened and transparent epoxy thermoset for advanced application. The data seems solid, and scientifically sound. The paper clearly indicated that the results obtained in this article can contribute to enhancing of toughening of epoxy/styrene/butadiene(SB)-based triblock copolymers by electron beam curing and thermal curing. However, the authors had better include the following points to make this manuscript more informative to readers. Therefore, this manuscript is recommended for the publication in materials after minor revision as below.
Response: The Authors would like to thank the reviewer for the evaluation of our manuscript. The reviewer’s comments were very helpful in improving the manuscript. The Authors have addressed the comments raised by the reviewer.
The author described that “Moreover, the increased toughness was an advantage of dual toughening of the blends via EB irradiation and post-thermal curing”: The EB curing condition such as irradiation dose is important one to enhance toughness of EP/eKraton blends. However, only two kinds of irradiation dose (20 and 50 kGy) were conducted to elucidate the effect of mechanical property of EP/eKraton blends. The reaction degree and crosslinking density of EP/eKraton blends as a function of irradiation dose was significant. However, there was not much explanation about that. So, the author should explain this clearly.
Response: The Authors agree. Definitely, the dose is the controlling parameter for electron induced reactions as well as size and distribution of microdomains in EP/eKraton blends (see Fig. 6). Consequently, the crosslinking behaviour of Kraton D 1101 was studied at a temperature of 215 °C in nitrogen atmosphere for different dose values (50, 100, 200 kGy) in [36]. In [36], the dominance of crosslinking reaction over chain scission was confirmed. Based on those results, the dose values of 20 and 50 kGy were precisely selected. The preparation of blends using a dose > 50 kGy were not successfully due to high values of gel content (> 89 %) of Kraton D 1101.

Reviewer 3 Report
Type of manuscript: Article
Title: A new way of toughening of thermoset by dual cured thermoplastic/thermosetting blend.
Journal: Materials
Corresponding Author: Uwe Gohs
Co-authors: Shankar P. Khatiwada , Ralf Lach, Gert Heinrich , Rameshwar Adhikari
This manuscript presents a study of toughening of thermoset by dual cured thermoplastic/thermosetting blend.
The manuscript contains some original work and the topic corresponds to the profile of Materials, the novelty are sufficient. Therefore the recommendation is: publish with required minor revision. Questions and comments are listed below: 1. In chapter 2. Experimental no grades of chemicals are given. 2. In the line 125 and 127, Figure 1: descriptions 1b and 1c are not good in the text! 3. Symbols are not visible 345, 347, 352, 654, 355, 367, 369, … etc.
Author Response
This manuscript presents a study of toughening of thermoset by dual curedthermoplastic/thermosetting blend. The manuscript contains some original work and the topiccorresponds to the profile of Materials, the novelty are sufficient.Therefore the recommendation is: publish with required minor revision.
Response:We would like to thank the reviewer for the evaluation of our work.
In chapter 2 Experimental no grades of chemicals are given.
Response: The authors don’t agree to the statement of the reviewer. Because diglycidyl ether of bisphenol A used as a matrix and 4,4-diaminodiphenylmethane (DDM) used as curing agent are provided by Sigma Aldrich no grades of chemicals are available. The styrene/butadiene-based linear triblock copolymer is a commercial product named as Kraton D 1101 (this information was already given).
In the line 125 and 127, Figure 1: descriptions 1b and 1c are not good in the text!
Response: Thank you for this comment.We changed to and 4,4-diaminodiphenylmethane (DDM), purchased from Sigma Aldrich, was used as curing agent (Figure 1c). The commercial styrene/butadiene-based linear triblock copolymer named as Kraton D 1101 (31 % by volume of styrene) was used as a modifier [34] (Figure 1b). The lab grade chemicals were used without any further purification.Because diglycidyl ether of bisphenol A and 4,4-diaminodiphenylmethane (DDM) are very usual chemicals for the synthesis of epoxy resins a detailed description of these chemicals was not provided.
Symbols are not visible 345, 347, 352, 654, 355, 367, 369, … etc.
Response: The authors agree. Now the symbols are visible.

Round 2
Reviewer 1 Report
All comments have been addressed.